

# A new species and new records of *Hymenopellis* and *Xerula* (Agaricales, Physalacriaceae) from China

Ya-jie Liu[1], Zheng-xiang Qi[1], You Li[1], Lei Yue[1], Gui-ping Zhao[1], Xin-yue Gui[1], Peng Dong[1], Yang Wang[1,2], Bo Zhang[1] and Xiao Li[1]

[1] Engineering Research Center of Edible and Medicinal Fungi, Ministry of Education, Jilin Agricultural University, Changchun, China

[2] College of Plant Protection, Shenyang Agricultural University, Shenyang, Ching

## ABSTRACT

*Hymenopellis* is the genus that exhibits the highest number of species within the *Xerula/Oudemansiella* complex. Numerous species of *Hymenopellis* demonstrate edibility, and some of these species have been domesticated and cultivated. During an extensive survey carried out in Henan and Jilin Provinces, China, a substantial quantity of *Hymenopellis* specimens was gathered as a component of the macrofungal resource inventory. Based on the findings of morphological and molecular phylogenetic studies, a new species, *Hymenopellis biyangensis*, has been identified. A new record species, *Hymenopellis altissima,* has been discovered in China. Additionally, two new record species, *Hymenopellis raphanipes* and *Xerula strigosa*, have been found in Henan Province. Internal transcribed spacer (ITS) and large subunit ribosomal (nrLSU) were used to establish a phylogeny for species identification. Detailed descriptions, field habitat maps and line drawings of these species are presented. The discussion focuses on the relationships between newly discovered species and other related taxa. Additionally, this study provides and a key to the documented species of *Hymenopellis* and *Xerula* found in China.

## INTRODUCTION

*Hymenopellis* RH Petersen is a genus of the *Xerula/Oudemansiella* complex which is characterized by a moist to glutinous pileus surface (*Petersen & Hughes, 2010*). Numerous species of *Hymenopellis* have been recorded as edible (*Wu et al., 2019*). One mushroom known as ''Heipijizong'' has been identified as *Hymenopellis raphanipes* and is extensively cultivated in China (*Hao et al., 2016*; *Niego et al., 2021*; *Sun et al., 2016*; *Xiao, He & Liu, 2022*).

The *Xerula/Oudemansiella* complex has a historical origin dating back to 1880 when Spegazzini first established the genus *Oudemansia* Speg. It was later renamed as *Oudemansiella* in 1881 by *Spegazzini (1881)*. In 1933, Maire established the genus *Xerula* (*Maire, 1933*). In the realm of classical taxonomy, the epitaxial and subordinate classification system of *Oudemansiella*, has been a subject of extensive debate among

Corresponding authors
Bo Zhang, zhangbofungi@126.com
Xiao Li, lxmogu@163.com

researchers, especially the relationship between *Oudemansiella*, *Mucidula*, *Mycenella* and *Xerula* (*Boursier, 1924*; *Clémençon, 1979*; *Dörfelt, 1979*; *Lange, 1914*; *Moser, 1955*; *Pegler & Young, 1986*; *Singer, 1964*). *Moser (1955)* proposed the *Oudemansiella* s.l., and merged *Xerula* and *Mucidula* into the genus *Oudemansiella*. *Singer (1964)*, *Singer (1986)* accepted this perspective and proposed the concept of the subtribe *Oudemansiellinae*, which include subg. *Oudemansiella* and subg. *Xerula*. *Clémençon (1979)* proposed a taxonomic system consisting of five subtribes and nine groups. This system was based on acceptance of *Xerula* as a subgenus of *Oudemansiella,* and *Mycenella* as a separate genus. However, *Pegler & Young (1986)* proceeded with the two subgenera treatments suggested by Singer and proposed a systematic treatment of two subgenera and five groups. *Dörfelt (1979)* retained *Oudemansiella* and *Xerula* as two separate genera, and some researchers also accepted this view (*Boekhout & Bas, 1986*; *Contu, 2000*; *Dörfelt, 1979*; *Halling & Mueller, 1999*; *Petersen, 2008*; *Petersen & Nagasawa, 2006*).

In the 21st century, researchers have demonstrated, based on the finding of morphological and molecular systematics, that *Oudemansiella* and *Xerula* are two independent genera. Furthermore, the numerous species that were previously classified under *Xerula* should now be reclassified under *Oudemansiella* (*Redhead, 1987*; *Wang et al., 2008*). *Zhang (2006)* showed that the *Oudemansiella* may be divided into three groups: sect. *Oudemansiella*, sect. *Dactylosporina* and sect. *Radicatae* based on molecular phylogenetic studies constructed from (ITS) and (nrLSU). *Wang et al. (2008)* redefined the genus *Xerula* s. str. and providing characterizations of the genus. *Yang et al. (2009)* excluded the narrowly defined genus *Xerula* from the genus *Oudemansiella* and categorized the genus *Oudemansiella* into four groups: sect. *Oudemansiella*, sect. *Mucidula*, sect. *Dactylosporina* and sect. *Radicatae*. *Petersen & Hughes (2010)* revised the taxonomic relationships of the *Xerula*/*Oudemansiella* complex based on morphological evidence and molecular phylogenetic results from ITS and nrLSU. They proposed 68 new taxa, including four new genera (*Hymenopellis*, *Paraxerula*, *Pointiculomyces*, *Protoxerula*), accepting the definition of the genus *Xerula* from *Yang et al. (2009)*, these results have been widely accepted and cited in subsequent studied (*He et al., 2019*; *Niego et al., 2021*; *Park et al., 2017*).

*Hymenopellis* species are widely distributed in Europe and North America (*Petersen & Hughes, 2010*), but they can also be found on other continents, such as Asia and Oceania (*Niego et al., 2021*; *Yang, 1993*). So far, there are 43 species records in Index Fungorum (http://www.indexfungorum.org) of which 12 species have been recorded in China (*Petersen & Hughes, 2010*; *Yang, 1993*; *Yao et al., 2020*).

During a survey of macrofungal resources in the Henan and Jilin Provinces of China, we collected some specimens of *Hymenopellis*. After conducting a comprehensive study that combins morphological and phylogenetic analysis, we the identification of a new species, a new record from China and two new records from Henan. These findings contribute to the overall species diversity within the taxon and provide valuable molecular data for further research in this field.
## MATERIALS AND METHODS

### Sampling and morphological analysis

The specimens were gathered from Biyang County, Henan Province and Shulan City, Jilin Province, China. Voucher specimens were deposited in the Herbarium of Mycology of Jilin Agricultural University (HMJAU). Macroscopic morphological characteristics are derived from field observations of fresh specimens, while microscopic morphology is examined using a light microscope. The primary reagents utilized in the analysis included a 5% solution of potassium hydroxide (KOH), a 1% solution of Congo red solution, and Meler's reagent solution. The colors of basidiomata were described using the color coding system developed by Kornerup and Wanscher's (*Kornerup & Wanscher, 1978*). The size of basidiospores is expressed as (a) b–c (d), where "a" is the minimum and "d" is the maximum, and 95% of the observed range falls between b–c. "Q" is the ratio of the length to width of the basidiospores, and "Q $\pm$ av" is the average Q $\pm$ standard deviation of all basidiospores (*Dong & Bau, 2022*; *Wang et al., 2022*).

### DNA extraction, PCR amplification, and sequencing

The total DNA was extracted from dried specimens using the NuClean Plant Genomic DNA kit (Kangwei Century Biotechnology Co., Ltd., Beijing, China). Primers for amplification of ITS were ITS 4 and ITS 5 (*He et al., 2023*; *Ward & Akrofi, 1994*), nrLSU were LROR and LR5 (*Kauserud & Schumacher, 2001*; *Wang & Yang, 2023*). The PCR program was as follows: pre-denaturation at 94 °C for 5 min; then followed by 35 cycles of denaturation at 94 °C for 50 s, annealing at 50 °C (ITS and nrLSU) for 50 s, and elongation at 72 °C for 70 s. Finally, a final elongation at 72 °C for 8 min was included. Then, the PCR products were sent to Biotechnology Co., Ltd. in Shanghai, China for sequencing.

### Phylogenetic analysis

The news sequences were uploaded on NCBI (http://www.ncbi.nlm.nih.gov). ITS and nrLSU sequences of related taxa were retrieved from GenBank and related articles (*Hao et al., 2016*; *Lin et al., 2021*; *Matheny et al., 2006*; *Niego et al., 2021*; *Petersen & Hughes, 2010*; *Qin et al., 2014*). The sequences obtained in this study, along with those of related taxa, are listed in Table 1. The ITS and nrLSU datasets include 109 sequences from 33 species of 7 genera within the Physalacriaceae family. *Flammulina yunnanensis* ZW Ge & Zhu L. Yang and *Flammulina velutipes* (Curtis) Singer were selected as outgroups for the phylogenetic analysis of the *Hymenopellis*, *Paraxerula americana* (Dörfelt) RH Petersen and *Paraxerula hongoi* (Dörfelt) RH Petersen were selected as outgroups for the phylogenetic analysis of *Xerula*.

The sequences from the two datasets were compared and manually modified separately using BIOEDIT (*Hall, Biosciences & Carlsbad, 2011*; *Ye & Bau, 2022*). PartitionFinder 2 was used to determine the optimal model scheme for two multi-locus datasets (*Lanfear et al., 2017*). Maximum parsimony analysis was applied to a combined dataset of two genera, following the approaches described by *Li, Zhao & Liu (2022)*. Phylogenetic analyses of the two genera were performed using the maximum likelihood (ML) and Bayesian inference (BI) methods, respectively. The maximum likelihood method (ML) was performed using

**Table 1  DNA sequence information for constructing phylogenetic trees.**

| Taxon | Voucher | Locality | GenBank | |
| --- | --- | --- | --- | --- |
| | | | **ITS** | **nrLSU** |
| *Flammulina velutipes* | TENN 56073 | London, England | AF030877 | HM005085 |
| *F. yunnanensis* | HKAS 32774 | China | DQ486704* | DQ457667* |
| *Gloiocephala aquatica* | CIEFAP 50 | Argentina | DQ097356* | DQ097343* |
| *G. menieri* | DAOM 170087 | Canada | DQ097358* | DQ097345* |
| ***Hymenopellis altitude*** | HMJAU 67050 | Henan, China | **OR035776** | **OR036095** |
| ***H. biyangensis*** | HMJAU 67043 | Henan, China | **OR035770** | **OR036089** |
| ***H. biyangensis*** | HMJAU 67044 | Henan, China | **OR035771** | **OR036090** |
| ***H. biyangensis*** | HMJAU 67045 | Henan, China | **OR035772** | **OR036091** |
| ***H. biyangensis*** | HMJAU 67046 | Henan, China | **OR035773** | **OR036092** |
| ***H. biyangensis*** | HMJAU 67047 | Henan, China | **OR035774** | **OR036093** |
| ***H. biyangensis*** | HMJAU 67048 | Henan, China | **OR035775** | **OR036094** |
| *H. chiangmaiae* | SFSU DED7661 | Malasia | HM011506 | HM005135 |
| *H. colensoi* | 12902,H2 | New Zealand | HM005140 | – |
| *H. colensoi* | 12902,H1 | New Zealand | HM005139 | – |
| *H. furfuracea* | TENN 59876 | TN, GSMNP, USA | GQ913367 | HM005126 |
| *H. furfuracea* | AFTOL-ID 538 | Massachusetts, USA | DQ494703 | AY691890 |
| *H. gigaspora* | TENN 50050 | NSW, Australia | GQ913359 | – |
| *H. gigaspora* | TENN 50056 | NSW, Australia | GQ913358 | – |
| *H. hispanica* | 05110401 (SEST) | Spain | – | HM005082 |
| *H. incognita* | TENN 58768 | TX, YSA | GQ913425 | HM005082 |
| *H. limonispora* | TENN 59438 | TN, Knox County, USA | GQ913406 | HM005133 |
| *H. limonispora* | TENN 61379 | TN, Knox County, USA | GQ913403 | – |
| *H. limonispora* | TFB 12913 | TN, Knox County, USA | – | HM005134 |
| *H. megalospora* | TENN 59556 | – | GQ913416 | – |
| *H. megalospora* | TENN 59556 | – | GQ913418 | – |
| *H. radicata* | TENN 62837 | Sweden | GQ913377 | HM005125 |
| *H. radicata* | TENN 60093 | Russia | GQ913393 | – |
| ***H. raphanipes*** | HMJAU 67039 | Henan, China | **OR035766** | **OR036085** |
| ***H. raphanipes*** | HMJAU 67051 | Henan, China | **OR035767** | **OR036086** |
| ***H. raphanipes*** | HMJAU 67041 | Henan, China | **OR035768** | **OR036087** |
| ***H. raphanipes*** | HMJAU 67042 | Henan, China | **OR035769** | **OR036088** |
| *H.raphanipes* | HKAS 95782 | Hunan, China | KX688236 | KX688263 |
| *H. raphanipes* | HKAS 95783 | Shandong, China | KX688238 | KX688265 |
| *H. sinapicolor* | TENN 56566 | Thailand | GQ913352 | HM005118 |
| *H. sinapicolor* | TENN 56566 | Thailand | GQ913353 | NG059449 |
| *Hymenopellis sp* | AGTN-2022b | Thailand | OP265165 | – |
| *Hymenopellis sp* | AGTN-2022b | Thailand | OP265164 | – |
| *H. trichofera* | MEL2293664 | Australia | GQ913354 | HM005129 |
| *H. vinocontusa* | TMI7669 | Japan | GQ913370 | – |

(*continued on next page*)

| Taxon | Voucher | Locality | GenBank | |
|---|---|---|---|---|
| | | | **ITS** | **nrLSU** |
| *Oudemansiella radicata var. australis* | HKAS47605 | Yunnan, China | AY961000[*] | AY960992[*] |
| *O. radicata var. australis* | HKAS47606 | Yunnan, China | AY961001[*] | AY960993[*] |
| *Paraxerula americana* | CLO4746 | New Mexico, USA | HM005142 | HM005094 |
| *P. americana* | DBG21746 | Colorado, USA | HM005143 | HM005093 |
| *P. ellipsospora* | HKAS 56261 | Yulong County, Yunnan, China | KF530557 | KF530567 |
| *P. hongoi* | HKAS 51985 | Hokkaido, Japan | KF530561 | KF530566 |
| *P. hongoi* | C 60612 | Japan | HM005144 | HM005095 |
| *Rhodotus asperior* | HKAS 56754 | Yingjiang County, Yunnan, China | KC179737 | KC179745 |
| *R. palmatus* | HMJAU 6872 | Antu County, Jilin, China | KC179742 | KC179752 |
| *Strobilurus albipilatus* | TENN52599 | Canada | GQ892804 | HM005089 |
| *S. conigenoides* | TENN61318 | North Carolina, USA | DQ097370 | HM005091 |
| *Xerula hispida* | TENN58745 | Costa Rica, San Jose | HM005164[*] | HM005098[*] |
| *X. melanotricha* | TFB11917 | Russia | HM005160[*] | HM005099[*] |
| *X. melanotricha* | LE10304 | Russia | HM005159[*] | |
| *X. melanotricha* | GMMueller7030 | – | AY665191[*] | AY804269[*] |
| *X. pudens* | Popa1969 | Germany | MF063189[*] | MF063124[*] |
| *X. pudens* | TFB11432 | Austrialia | HM005154 | HM005097 |
| *X. pudens* | C 63308 | Spain | HM005155 | – |
| *X. sinopudens* | Feng266 | Xishuangbanna, Yunnan, China | KF530551 | KF530571 |
| *X. sinopudens* | ZRL20151504 | – | LT716059[*] | KY418875[*] |
| **X. strigosa** | **HMJAU67049** | Henan, China | **OR030918** | **OR030419** |
| *X. strigosa* | Zhu528 | Cheng county, Gansu China | KF530556[*] | KF530569[*] |
| *X. strigosa* | Zhu513 | Cheng county, Gansu, China | KF530555 | KF530568 |

**Notes.**
[*]Sequence retrieved from GenBank. The newly generated sequences in this study are indicated in bold.

IQTree 1.6.8 (*Nguyen et al., 2015*), with ultrafast bootstrapping and 5,000 repetitions. The Bayesian inference (BI) was performed using MrBayes 3.2.6 (*Deng et al., 2022*; *Ronquist et al., 2012*) in PhyloSuite 1.2.2 (*Zhang et al., 2020*), running for 2,000,000 generations. The initial 25% of the sampled data was discarded as burn-ins.

## Nomenclature

"The electronic version of this article in Portable Document Format (PDF) will represent a published work according to the International Code of Nomenclature for algae, fungi, and plants, and hence the new names contained in the electronic version are effectively published under that Code from the electronic edition alone. In addition, new names contained in this work have been submitted to MycoBank from where they will be made available to the Global Names Index. The unique MycoBank number can be resolved and the associated information viewed through any standard web browser by appending the MycoBank number contained in this publication to the prefix "http://www.mycobank.org/MycoTaxo.aspx?Link=T&Rec=". The online version of this work is

## RESULTS

### Molecular phylogeny

The two-locus gene dataset (ITS + LSU) of *Hymeopellis* contains 51 sequences and 1,770 aligned characters in length. Out of these, 1,242 characters are constant, 439 characters are parsimony-informative, and 89 characters are variable and parsimony-uninformative. For the multi-locus data set of *Hymenopellis*, the optimal model for ITS was HKY + F + I + G4 (*Liao et al., 2022*), and the optimal model for nrLSU was K2P + I (*Ripplinger & Sullivan, 2008*). Because the phylogenetic trees generated from the combined dataset using ML and BI analyses exhibited nearly identical topology, we have chosen to display only the ML tree (Fig. 1). In the phylogenetic tree, *H. biyangensis* and *H. altissima* are belonged to the genus *Hymenopellis*, and each forms a well-supported branch (BP = 99, PP = 1 and BP = 99, PP = 1).

*Hymenopellis raphanipes* collected from Biyang County, China, is found on the same branch as *Hymenopellis raphanipes* distributed in two other locations (Fig. 1), and it received strong support (BP = 100, PP = 1).

The two-locus gene dataset (ITS + LSU) of *Xerula* contains 16 sequences and 1,676 aligned characters in length. Out of these, 1,395 characters are constant, 202 characters are parsimony-informative, and 79 characters are variable and parsimony-uninformative. For the multi-locus data set of *Xerula*, the optimal model for ITS was HKY + G (*Liu et al., 2017*), and the optimal model for nrLSU was GTR + I (*Posada, 2008*). In the phylogenetic tree, *Xerula strigosa* collected from Biyang County, China was located in the same clade as *X. strigosa* from Yunnan (Fig. 2), which received high support (BP = 100, PP = 1).

### Taxonomy

***Hymenopellis biyangensis* Y.J. Liu, B. Zhang & Xiao Li sp. nov.**
MycoBank No: 848957
Figs. 3A–3B and 4

**Etymology.** "biyangensis" refers to its type locality Biyang County.
**Holotype.** China, Henan Province, Zhumadian City, Biyang County, WanFenSi, 113°36′40″E, 32°51′45″N, 11 July, 2021 (HMJAU67048).
**Description.** Basidiomata moderate-to-large sized. Pileus 30–105 mm in diam, surface dry to viscid, sticky when wet, subumbonate, convex to plano-concave, darker in center, brownish gray (10E2) or reddish brown (8D5), outward brownish orange (6C3) to light brown (7D4), yellowish grey (3B2), brown (7E4) when young, slightly to strongly radially wrinkled from subumbonate onto margin (at margin occasionally net-like), sometimes with short inconspicuous stripes on margin. Context white (1A1), not discolored when cut. Lamellae adnate with decurrent tooth, white (1A1), sometimes with dark brown

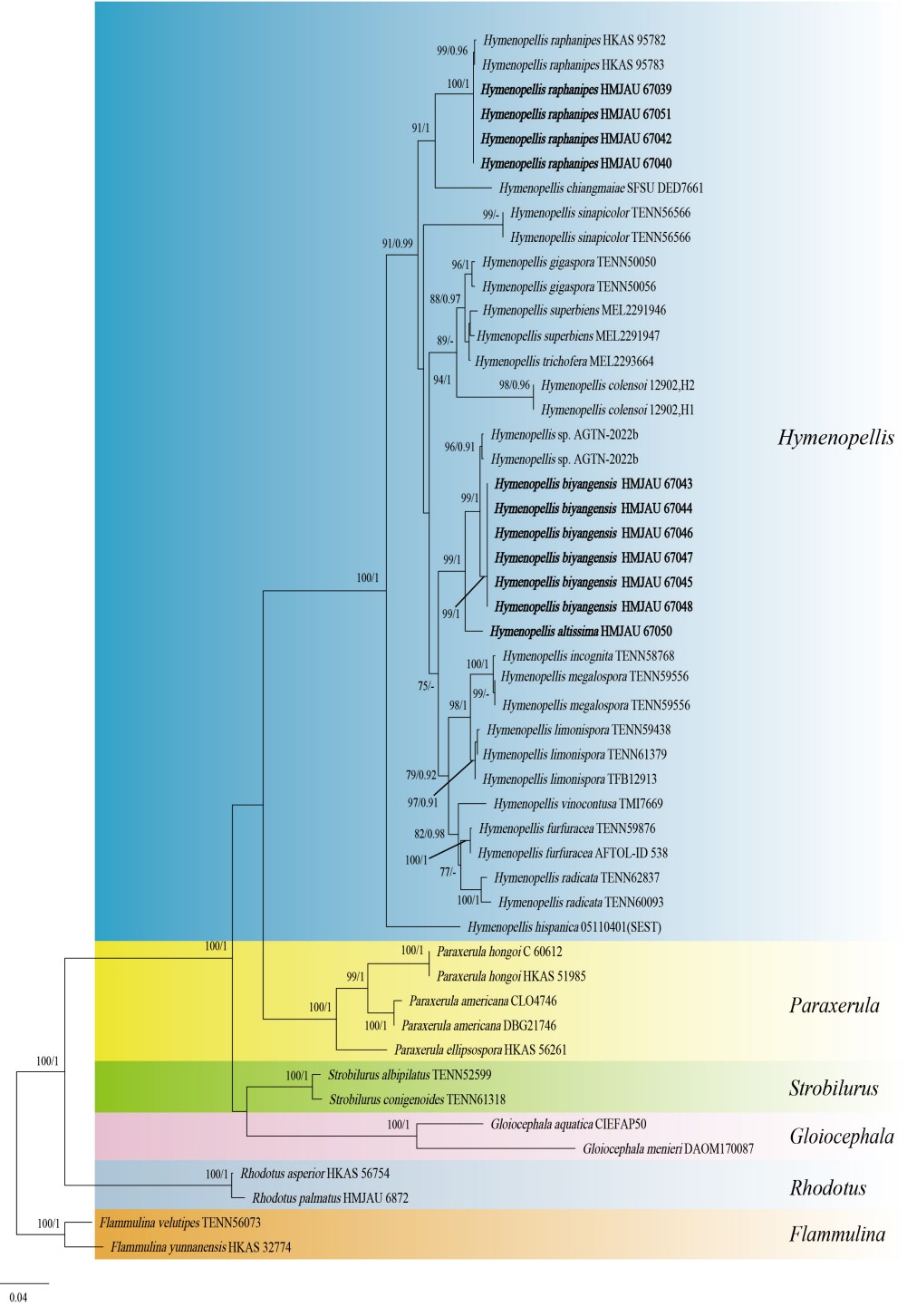

0.04

**Figure 1 Phylogenetic tree constructed from the combined ITS and nrLSU dataset using ML methods.** The two values of the internal node represent Maximum Likelihood bootstrap (MLBP > 70%)/Bayesian posterior probability (BIPP > 90%). Our species sequences are marked in bold.

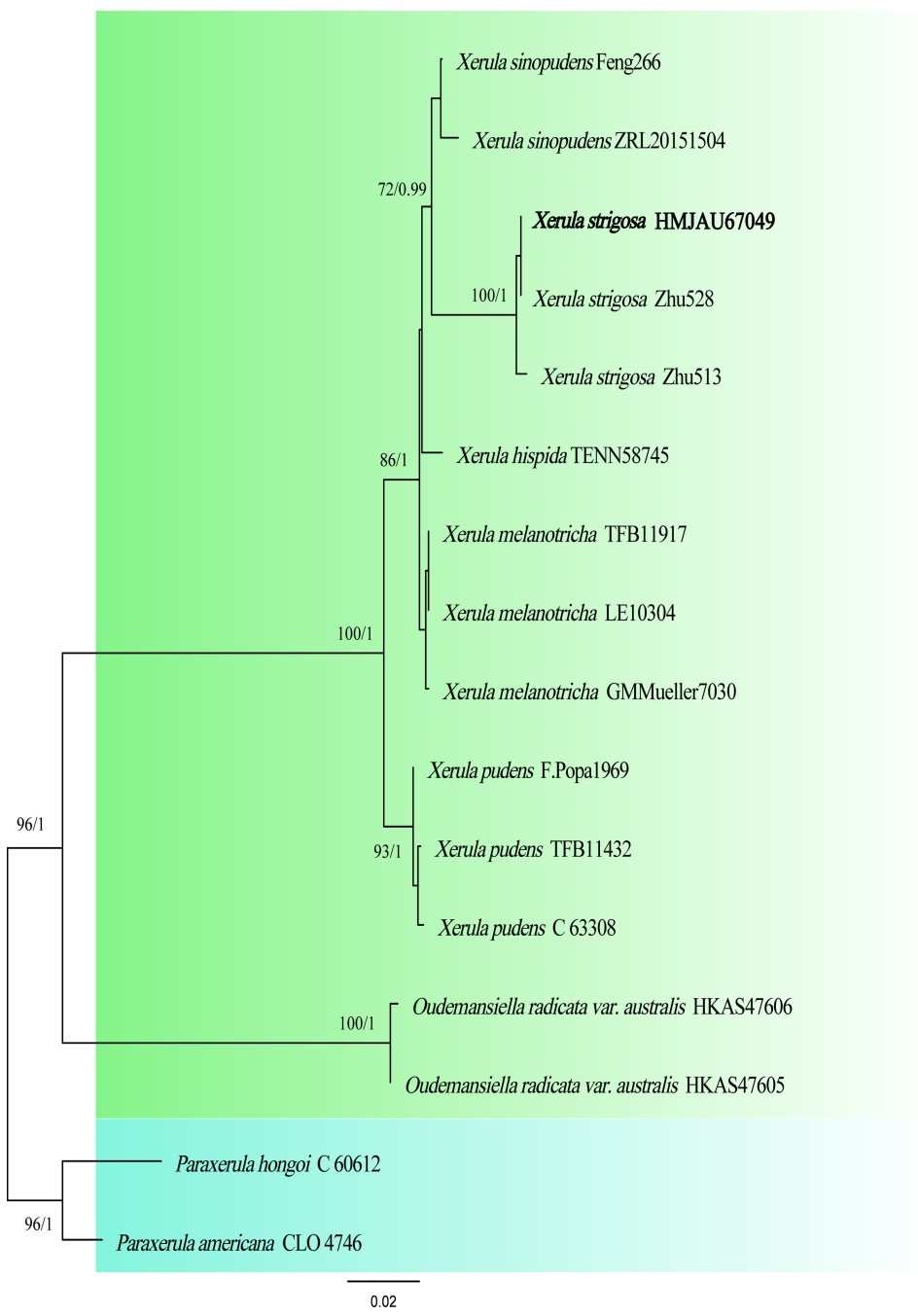

**Figure 2** **Phylogenetic tree constructed from the combined ITS and nrLSU dataset using ML methods.** The two values of the internal node represent Maximum Likelihood bootstrap (MLBP > 70%) / Bayesian posterior probability (BIPP > 90%). Our species sequences are marked in bold.

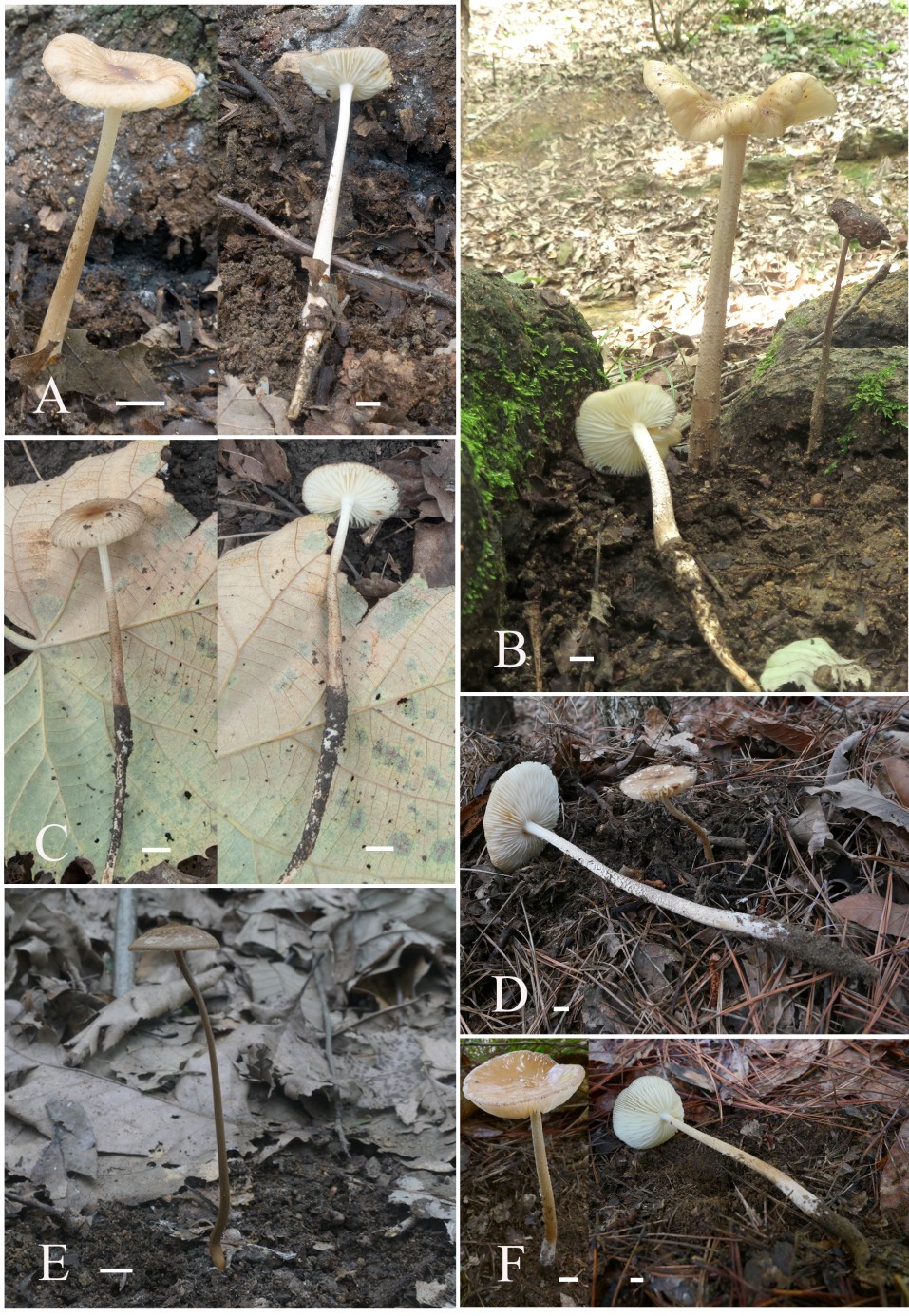

**Figure 3** **Basidiomata of *Hymenopellis* and *Xerula*.** (A, B) *Hymenopellis biyangensis* (A HMJAU67045, B HMJAU67048) (C) *Hymenopellis altissima* (HMJAU67050) (D, F) *Hymenopellis raphanipes* (D HM-JAU67041, F HMJAU67051) (E) *Xerula strigosa* (HMJAU67049). Scale bars: 1 cm.

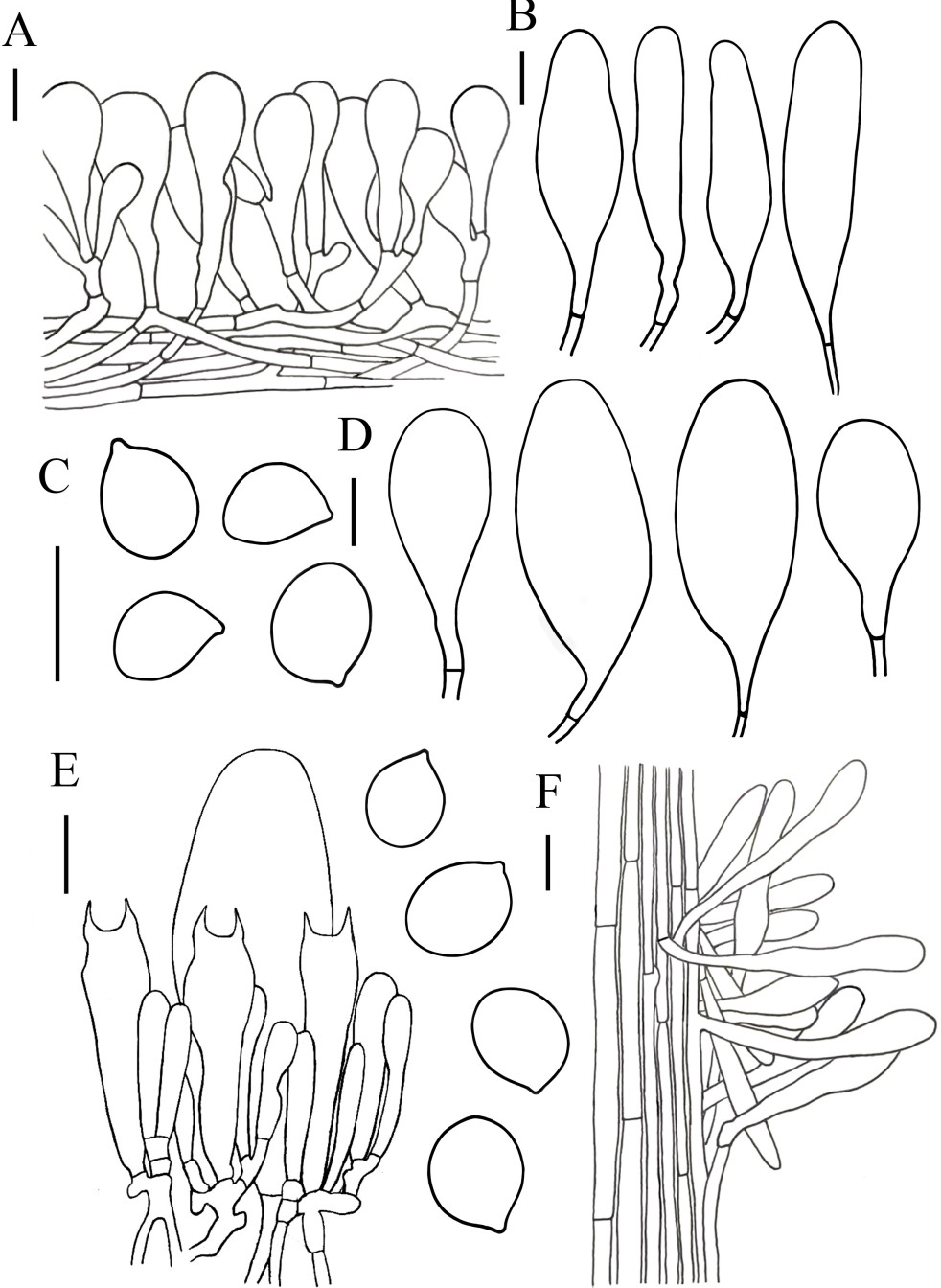

**Figure 4** **Microscopic features of *Hymenopellis biyangensis*..** (A) Pileipellis (B) Cheilocystidia (C) Basidiospores (D) Pleurocystidia (E) Hymenium (F) Caulocystidia. Scale bars: 20 μm.

(8F8) spots, subdistant, in 3–4 tiers, thick, 2–9 mm deep. Stipe 75–100 × 5–14 mm, hollow, subcylindrical, tapering upward, white (1A1) to brown (7E4) from the pileus to downwards, with ciliate squamulose. Pseudorhiza at least 62 mm long, 10–14 mm broad, tapering downward, brownish orange (6C4, 6C5) to light brown(6D5).

Lamelar trama subregular, composed of 3–30 μm diameter, hyaline hyphae, without clamp connection. Basidia 40–70 × 9–19 μm, clavate, with 2-spored, rarely 4-spored, sterigmata 5–14 μm long, without clamp connection. Basidiospores (13) 14–20 (20.5) × (8.5) 11–15 (17) μm, [Q = (1.07) 1.13–1.52 (1.57), $Q_m = 1.28 \pm 0.12$], subglobosus, ellipsoid to broadly ellipsoid, hyaline in 5% KOH, smooth, inamyloid. Pleurocystidia 62–143 × 23–56 μm, pedicellate, broadly clavate, paddle-sha ped to obovoid, hyaline or with yellow intracellular pigment. Cheilocystidia abundant, 48–137 × 10–36 μm, pedicellate, clavate, broadly utriform, hyaline. Stipe trama monomitic, composed of parallel hyphae 5–30 μm wide, hyaline. Stipitipellis 3–12 μm, composed of flavidus parallel hyphae, clamp connection rare. Caulocystidia 36–119 × 7–17 μm, narrowly clavate. Pileipellis a hymeniderm of 100–160 μm thick. Pileipellis over disc and near pileus margin similar, constructed by clavate, broadly clavate to sphaeropedunculate elements, 20–97 × 12–36 μm, hyaline or with brown intracellular pigment. Pileal hairs absent.

**Habitat.** Solitary or scattered on the soil of *Quercus acutissima* forest or *Quercus acutissima* and *Castanea mollissima* mixed forest.

**Distribution.** Henan Province, China

**Additional specimens examined.** China, Henan Province, Zhumadian City, Biyang County, Minzhuan Forest Farm, 32°52′18″N, 113°36′25″E, 10 July 2021. ZhengXiang Qi (HMJAU67043, HMJAU67044), GuiPing Zhao (HMJAU67047); China, Henan Province, Zhumadian City, Biyang County, WanFenSi, 32°51′45″N, 113°36′40″E, 11 July 2021. ZhengXiang Qi (HMJAU67045, HUJAU67046).

**Notes.** *Hymenopellis biyangensis* is characterized by its pileipellis lacks pileal hairs, broadly clavate, paddle-shaped to obovoid pleurocystidia and 2-spored basidia. *Hymenopellis biyangensis* is morphologically similar to *H. furfuracea*, but *H. furfuracea* differs in utriniform-pedicellate to narrowly ten pin-shaped (fusiform-subcapitulate) pleurocystidia (*Petersen & Hughes, 2010*; *Yang, 1993*). *Hymenopellis biyangensis* is also similar to *H. vinocontusa*, but the latter lacks caulocystidia (*Petersen & Hughes, 2010*).

*Hymenopellis biyangensis* have 2-spored basidia. According to *Petersen & Hughes (2010)* description of the 2-spored basidia species, *H. biyangensis* is similar to *H. incognita* (basionym: *H. incognita* f. *bispora*), but the latter has solid stipe and utriform, commonly submammilate pleurocystidia. *Hymenopellis biyangensis* is also similar to *H. radicata* (basionym: *H. radicata* var. *bispora*), but *H. radicata* differs in pileus margin without stripes, broadly fusiform with bluntly rounded apically pleurocystidia and obpyriform pileipellis elements.

*Hymenopellis biyangensis* have paddle-shaped to obovoid pleurocystidia, which is relatively uncommon in the genus. In this terms, *H. biyangensis* is similar to *H. colensoi*, but *H. colensoi* differs in pileipellis has pileal hairs and globose to subglobose basidiospores (*H. colensoi* 12–16 × 12.5–13.5 μm *vs. H. biyangensis* 14–20 × 11–15 μm) (*Petersen & Hughes, 2010*). *Hymenopellis biyangensis* is also similar to *H. trichofera*, but the latter pseudorhiza has white tomentum and the pileipellis has pileal hairs (*Petersen & Hughes, 2010*).

*Hymenopellis altissima* (Massee) RH Petersen
Basionym: *Collybia altissima* Massee, Bull. Misc. Inf., Kew(10): 358 (1914)
= *Oudemansiella altissima* (Massee) Corner, Gdns' Bull., Singapore 46 (1): 56 (1994)
Figs. 3C and 5

**Description.** Basidiomata small sized. Pileus diameter 35 mm, plano-convex, greyish brown (7D3) to reddish brown (8E6), darker in the middle, strongly radially wrinkled from middle onto limb, stripes on margin. Context white (1A1), not discolored when cut. Lamellae white (1A1), adnate, disatant, thick, in 4 tiers. Stipe 63 × 3.8 mm, hollow, subcylindrical, gradually thickening down to the pseudorhiza, white (1A1) to brown (7E4) from the pileus to downwards, with ciliate squamulose. Pseudorhiza at least 77 mm, 5 mm wide, slightly tapered downward, white (1A1) to light brown (7D4).

Lamellar trama regular, composed of hyphae 3–23 μm diameter, without clamp connection. Basidiospores (13) 14–17.5 (18) × (12.5) 13–16 (16.5) μm, [Q = (1) 1.03–1.21 (1.29), $Q_m$ = 1.10 ± 0.06], globose, subglobose, a few broadly ellipsoid, hyaline, smooth, thin-walled, inamyloid. Basidia 47–63 × 14–17 μm, clavate, 2-spored, sterigmata 5–9 μm long, without clamp connection. Pleurocystidia 90–165 × 22–39 μm, pedicellate, fusiform, obviously capitulates, hyaline or with yellow pigment, without clamp connection. Cheilocystidia abundant, 62–127 × 13–32 μm, pedicellate, clavate, narrowly utriform, hyaline, thin-walled. Stipitipellis composed of vertically arranged, hyaline, slightly thick-walled (≤1 μm), hyphae 3–16 μm broad. Caulocystidia 54–156 × 10–22 μm, clavate to narrowly utriform, thinning at the end, hyaline. Pileipellis an ixohymeniderm of 100–160 μm thick. Pileipellis over disc constructed by clavate to broadly clavate elements, 43–104 × 19–30 μm, hyaline or with brown intracellular pigment; pileipellis near margin constructed by sphaeropedunculate to clavate elements, 39–85 × 21–30 μm, hyaline or with brown intracellular pigment. Pileal hairs absent.

**Habitat.** Growing on the ground in a mixed forest of *Quercus mongolica* and *Tilia tuan*.

**Distribution.** China, Indonesia, Japan, Malaysia, Russian Federation, Singapore (*Petersen & Hughes, 2010*).

**Specimens examined.** China, Jilin Province, Shulan City, Jiulongshan Forest Park. 3 August 2022, Lei Yue (HMJAU67050).

**Notes.** *Hymenopellis altissima* is characterized by strongly radially wrinkled from pileus middle onto limb, basidiospores globose, subglobose, pleurocystidia fusiform, obviously capitulated, without clamp connection. Our specimens have longer pseudorhiza (77 mm) differs from those described by other authors, who describe it as up to 16 mm (*Petersen & Hughes, 2010*) or 160 mm (*Corner, 1994*). Morphologically, *H. altissima* is relatively similar to *H. vinocontusa*, but *H. vinocontusa* differs in its ellipsoidal, ovate to sublimoniform basidiospores and fusiform-capitulated cheilocystidia (*Petersen & Hughes, 2010*). In addition, *H. vinocontusa* has 4-spored basidia, while *H. altissima* has 2-spored basidia.

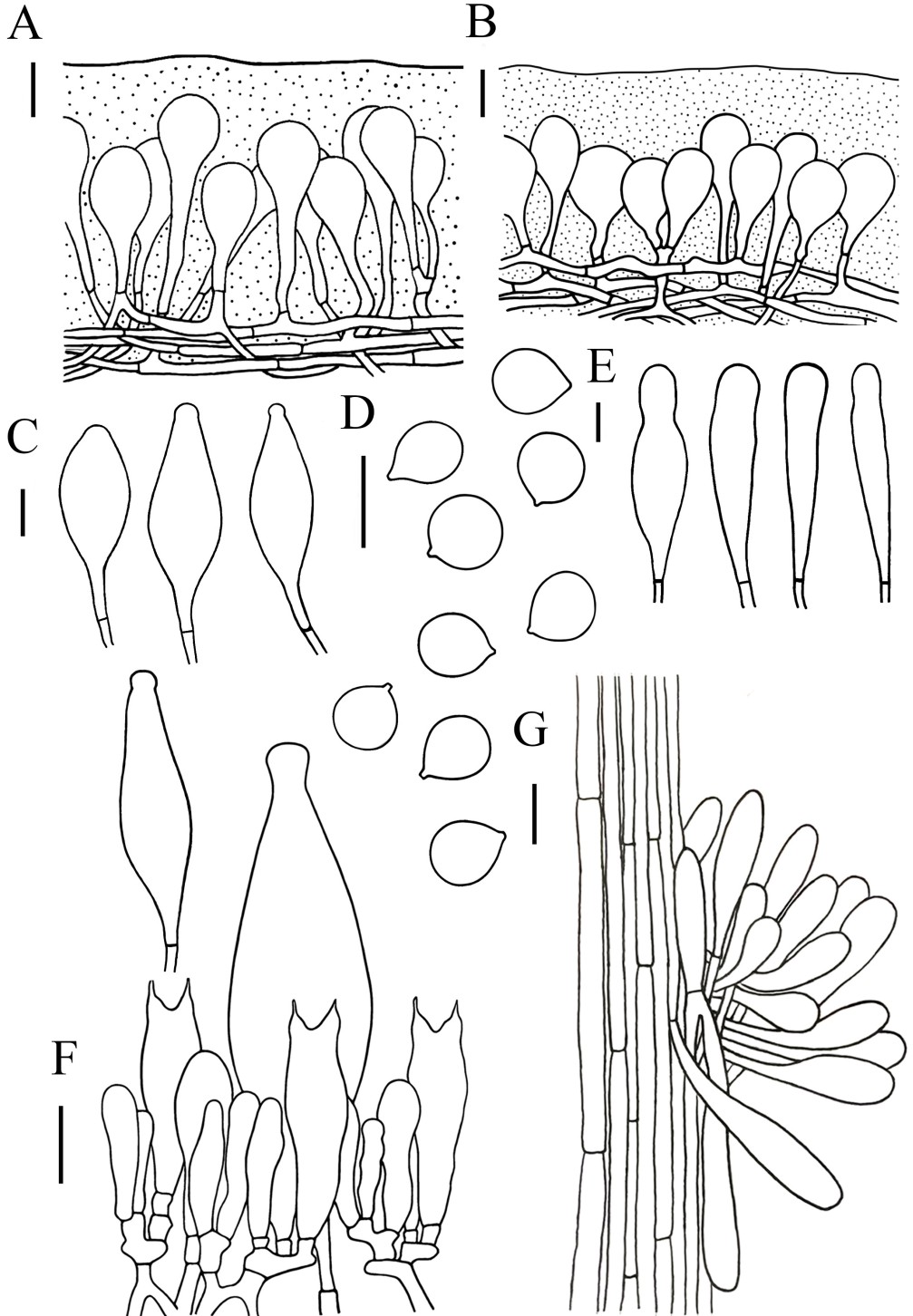

**Figure 5  Microscopic features of *Hymenopellis altissima.*.** (A) Pileipellis over disc (B) Pileipellis near margin (C) Pleurocystidia (D) Basidiospores (E) Cheilocystidia (F) Hymenium (G) Caulocystidia. Scale bars: 20 μm.

*Hymenopellis raphanipes* (Berk.) R.H. Petersen

Basionym: *Agaricus raphanipes* Berk., Hooker's J. Bot. Kew Gard. Misc. 2: 48, 1850

Figs. 3D, 3F and 6

**Description.** Basidiomata small-to-large sized. Pileus 23–95 mm in diam, initially hemispherical, later nearly expansus, slightly convex or concave, margin or upturned, margin dehiscence, greyish brown (6D2), brownish orange (7C3), light brown (6D4), brown (7E8) to dark brown (7F8), smooth, sometimes with reticulate veins near the margin. Context white (1A1), not discolored when cut. Lamellae adnate to sinuate or slightly decurrent, subdistant, in 3–4 tiers, thick, 6–12 mm deep, white (1A1) to cream (1B1), sometimes with brown(6E5, 7E6) spots. Stipe 64–200 × 4–12 mm, solid, subcylindrical, tapering upward, white (1A1) to grey (10C1, 13D1–16D1), surface densely covered with brown (7E6) felted squamulose but nearly white at apex. Pseudorhiza 35–75 × 8–13 mm, slightly tapered downward, white (1A1) to brown (6D7).

Lamellar trama regular, composed of 3–20 μm diameter, hyaline hyphae, without clamp connection. Basidia 32–75 × 9–19 μm, clavate, 2-spored, rarely 4-spored, sterigmata 5–12 μm long, thin-walled, hyaline. Basidiospores (14) 16–22 (23) × (7) 10–14 (15) μm, [$Q = (1.25)\ 1.36–1.73\ (1.75)$, $Q_m = 1.52 \pm 0.12$], ovoid, ellipsoid to broadly ellipsoid, hyaline, inamyloid, smooth. Pleurocystidia 66–150 × 20–48 μm, pedicellate, fusiform with prominent capitate apex, slightly thick-walled, hyaline. Cheilocystidia 52–274 × 14–50 μm, pedicellate, fusiform, narrowly clavate to clavate, thin-walled, hyaline. Stipitipellis consists of hyaline hyphae arranged longitudinally with a diameter of 3–20 μm, without clamp connection. Caulocystidia 35–206 × 7–19 μm, clavate, cylindrical, hyaline. Pileipellis a hymeniderm with some extended pileal hairs. Pileipellis over disc and near pileus margin similar, constructed by clavate, broadly clavate to sphaeropedunculate elements, 22–70 × 8–24 μm, hyaline or with brown intracellular pigment. Pileal hairs 65–181 × 9–22 μm, narrowly utriform to subcylindrical, thin-walled, hyaline.

**Habitat.** Solitary or scattered on the soil of mixed forests of *Quercus robur* and *Torrey pine*.

**Distribution.** Australia, China, India, Japan, Thailand (*Petersen & Hughes, 2010*).

**Specimens examined.** China, Henan Province, Zhumadian City, Biyang County, Tongshan Lake Forest Park, 32° 45′31″N, 113°30′51″E, 9 July 2022, YaJie Liu (HMJAU67039), 13 July 2022, YaJie Liu (HMJAU67040), 23 July 2022, YaJie Liu (HMJAU67051), 27 July 2022, YaJie Liu (HMJAU67042).

**Notes.** *Hymenopellis raphanipes* is characterized by small-to-large sized basidiomata, ovoid, ellipsoid to broadly ellipsoid basidiospore and fusiforms with prominent capitate apex pleurocystidia. *Hymenopellis raphanipes* is morphologically similar to *H. radicata*, but *H. radicata* differs in its glabrous stipe and the absent of pileal hairs (*Hao et al., 2016*; *Petersen & Hughes, 2010*). *Hymenopellis raphanipes* is also similar to *H. furfuracea*, but *H. furfuracea* differs in its hollow stipe and utriform-pedicellated to ten pin-shaped pleurocystidia (*Hao et al., 2016*; *Petersen & Hughes, 2010*).

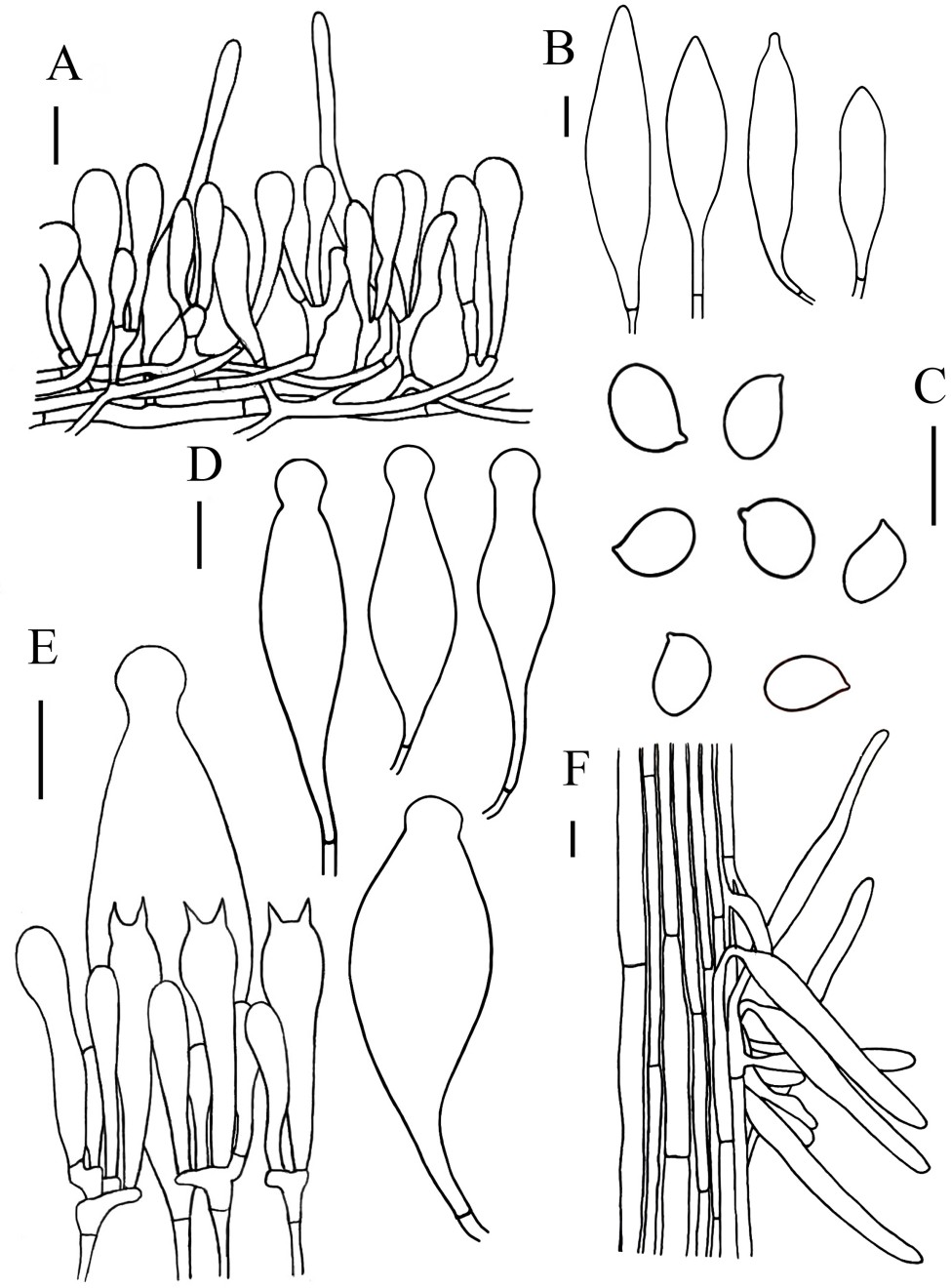

**Figure 6** **Microscopic features of *Hymenopellis raphanipes*..** (A) Pileipellis (B) Cheilocystidia (C) Basidiospores (D) Pleurocystidia (E) Hymenium (F) Caulocystidia. Scale bars: 20 μm.

***Xerula strigosa*** Zhu L. Yang, L. Wang & G.M. Muell. 2008
Figs. 3E and 7

**Description.** Basidiomata small sized. Pileus 15–30 mm in diam, convexus to plano-convexus, yellowish brown (5E5) to dark brown (6F7), dry, surface covered with brown setae, smooth at the margin, without stripes. Context white(1A1). Lamellae free to adnate, subdistant, white (1A1), with lamellulae, in 3 tiers, up to four mm deep. Stipe 100–125 × 2.5–3 mm, solid, cylindrical or slightly tapered upward, reddish brown (8E8) to dark brown (7F4), paler towards the apex, surface covered with setae of the same color as the stipe. Pseudorhiza 36–42 mm long, slightly tapered downward.

Lamellar trama subregular, composed of 3–33 μm diameter, hyaline. Basidiospores (11) 12–16 (17) × (8) 9–12 (13) μm, [Q = (1.00) 1.09–1.52 (1.63), $Q_m = 1.30 \pm 0.15$], subglobosus, boradly ellipsoid to ellipsoid, hyaline. Basidia 27–67 × 13–18 μm, clavate, 4-spored, rarely 2-spored, sterigmata 5–9 μm long, without clamp connection at base. Pleurocystidia 78–135 × 18–33 μm, narrowly utriform, capitate to subcapitate at the top, often with yellowish crystalline deposits, without clamp connection. Lamellar edge fertile, with scattered cheilocystidia. Cheilocystidia 75–106 × 18–27 μm, narrowly utriform, hyaline, thick-walled (wall-2 μm thick), without clamp connection. Stipitipellis composed of vertically arranged, slightly yellow, hyaline, slightly thick-walled (≤ 1μm), hyphae 3–11 μm broad. Caulocystidia similar to pileosetae, 40–236 × 3–12 μm, lanceolate, thick-walled (≤ 2μm), yellowish brown. Pileipellis a hymeniderm with pileosetae. Pileipellis over disc constructed by narrowly clavate to clavate elements, 51–105 × 13–20 μm, hyaline or with light brown intracellular pigment; pileipellis at pileus margin constructed by subfusiform, pyriform or sphaeropedunculate elements, 43–63 × 12–18 μm, hyaline or with light brown intracellular pigment. Pileosetae 93–540 × 8–18 μm, lanceolate, thick-walled (wall 1–3 μm thick), yellowish brown.

**Habitat.** Growing on the ground in mixed broad-leaved forests dominated by *Quercus robur*.

**Distribution.** China, Pakistan (*Krisai-Greilhuber et al., 2017*).

**Specimens examined.** China, Henan Province, Zhumadian City, Biyang County, Minzhuan Forest Farm, 32°52′18″N, 113°36′25″E, 10 July 2021. ZhengXiang Qi (HMJAU67049).

**Notes.** *Xerula strigosa* is characterized by boradly ellipsoid to ellipsoid basidiospores, pleurocystidia narrowly utriform, capitate to subcapitate at the top and lanceolated pileocystidia. *Xerula strigosa* closely resembles *X. pudens* (*Wang et al., 2008*), compared with *X. pudens*, *X. trigosa* exhibits longer spores and thin-walled apex of pleurocystidia. *Xerula strigosa* also strongly resembles the *X. hispida*, but *X. hispida* differs in its 2-spored (rarely 4) basidia and pleurocystidia without a capitate apex.

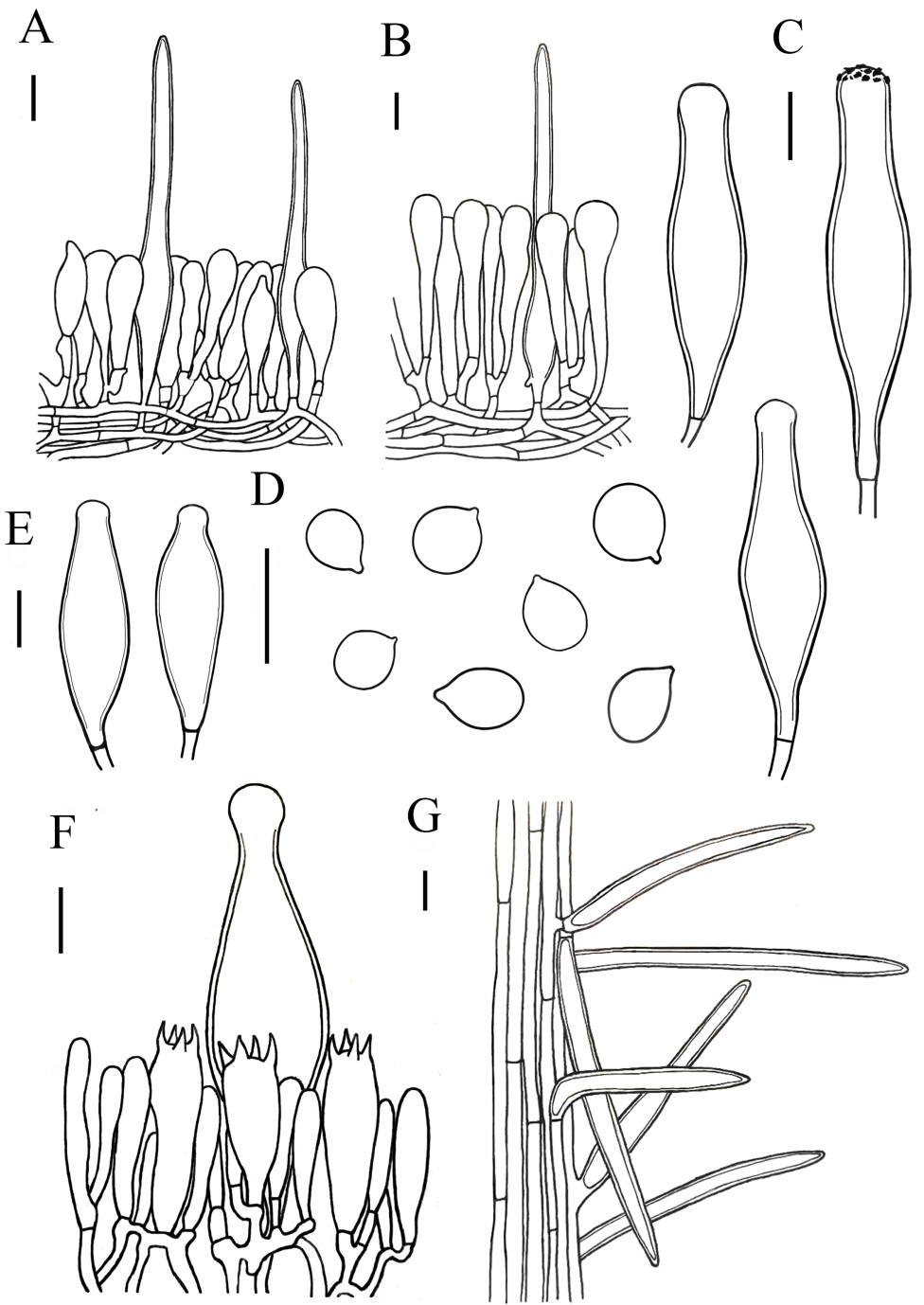

**Figure 7** **Microscopic features of *Xerula strigosa*..** (A) Pileipellis near margin (B) Pileipellis over disc (C) Pleurocystidia (D) Basidiospores (E) Cheilocystidia (F) Hymenium (G) Caulocystidia. Scale bars: 20 μm.

A key to China species of *Hymenopellis*

1. Pileipellis with pileal hairs....................................................................................2
– Pileipellis without pileal hairs..............................................................................8
2. Context white beneath pileipellis and above lamellae.........................................3
– Context fuliginous to avellaneous beneath pileipellis and above lamellae...***H. colensoi***
3. Basidiospores amygdaliform....................................................***H. amygdaliformis***
– Basidiospores broadly ellipsoid, ovatus to sublimoniform.........................................4
4. Cheilocystidia narrowly clavate to clavate, fusiform..........................................5
– Cheilocystidia fusiform with flagelliform apex............................***H. hygrophoroides***
5. Pleurocystidia capitate or capitulate apex..........................................................6
– Pleurocystidia without capitate or capitulate apex.................................................7
6. Pleurocystidia utriform....................................................................***H. furfuracea***
– Pleurocystidia fusiform ...................................................................***H. raphanipes***
7. Pileus subumbonate, margin of the pileus with sripes.....................***H. megalospora***
– Pileus subumbonate, margin of the pileus without stripes............................***H. bispora***
8. Stipe with annulate veil.......................................................................***H. velata***
– Stipe without annulate veil..................................................................................9
9. Basidia 2-spored...............................................................................................10
– Basidia 4-spored...............................................................................................11
10. Pleurocystidia paddle-shaped, obovoid.................................................***H. biyangensis***
– Pleurocystidia fusiform, obviously capitulates..........................................***H. altissima***
11. Caulocystidia absent.......................................................................***H. vinocontusa***
– Caulocystidia exists..........................................................................................12
12. Cheilocystidia broadly cylindrical.....................................................................13
– Cheilocystidia narawly utriform, fusiform, narrowly clavate to clavate.....................14
13. Stipe longitudinally lined, usually twisted, sometimes reluctantly reddish brown in spots. or suffused below..................................................................................***H. radicata***
– Stipe appearing minutely laccate as though viscid, with darker amorphous patches…….….. ...................................................................................***H. orientalis***
14. Basidiospores globose to subglobose……………….. …................... ***H. japonica***
– Basidiospores subovate to sublimoniform…………………….... ***H. aureocystidiata***

A key to China species of *Xerula*

1. Basidia 2-spored, basidiospores globose, subglobose to ovoid ….….….***Xerula hispida***
– Basidia 4-spored, basidiospores subglobose to broadly ellipsoid.….............................2
2. Pleurocystidia fusiform, often with crystalline deposited apex…………..…......…....3
– Pleurocystidia fusiform, without crystalline deposits…………..…........***X. sinopudens***
3. Clamp connections common…………………………….………..............***X. puden***
– Clamp connections absent…………………………….………..................……***X. strigosa***

## DISCUSSION

In this study, a new species, *Hymenopellis biyangensis*, was discovered, along with a new record species of *H. altissima* for China. Additionally, two new record species, *H. raphanipes* and *X. strigos a* were found in Henan province through the use of morphology and molecular phylogeny.

In the phylogenetic analysis, the new species *H. biyangensis* formed an independent clade and was found to be a sister group with *H. altissima*, which received high support (BP = 99, PP = 1) (Fig. 1). However, the pileipellis of *H. altissima* has an ixohymeniderma while *H. biyangensis* does not. In addition, *H. biyangensis* and *H. altissima* can be distinguished by their pleurocystidia and basidiospores. *H. biyangensis* have paddle-shaped, obovoid pleurocystidia and ellipsoid to broadly ellipsoid basidiospores, while *H. altissima* has fusiform, obviously capitulates pleurocystidia and globose to subglobose basidiospores.

Pleurocystidia of *Hymenopellis* species are mostly described as "utriform", "jar-shaped" and "ten pin-shaped" (*Petersen & Hughes, 2010*), while *H. biyangensis* has broadly clavate, paddle-shaped to obovoid pleurocystidia. In addition, both *H. biyangensis* and *H. altissima* are relatively rare 2-spore species in the genus *Hymenopellis*. Among the 2-spored species, the basidiospores of *H. altissima* are subglobose to globose different from the other species. *Hymenopellis biyangensis*, *H. altissima* and *H. raphanipes* are similar in habitat, growing on the forest dominated by Fagaceae. In the description by *Hao (2016)*, *Petersen & Hughes (2010)*, thirteen species of *Hymenopellis* that were found to grow in Fagaceae forests, particularly in *Fagus* and *Quercus*. This suggests that there may be a relationship between the growth of species in this genus and Fagaceae, which should be further studied.

*Hymenopellis* is a paraphyletic taxon (*Vellinga, 2010*) and represents a group of morphologically similar taxa that were previously distributed in several sections of *Oudemansiella* (*Petersen & Hughes, 2010*). Therefore, some researchers have also questioned the establishment of *Hymenopellis*. *Hao (2016)* proposed a new taxonomic system in which *Oudemansiella* is recognized as a highly supported monophyletic lineage, and should be treated as a single genus. A new systematic arrangement with three sections, namely, sect. *Oudemansiella*, sect. *Dactylosporina* and sect. *Radicatae* has been proposed; *Hymenopellis* should be subsumed into the genus *Oudemansiella* (*Hao, 2016*). Therefore, more researches on this taxon is still needed in the future.

Four species of *Xerula*—*Xerula hispida*, *X. pudens*, *X. sinopudens* and *X. strigosa*—have been previously reported in China (*Mueller et al., 2001*; *Wang et al., 2008*; *Yao et al., 2020*). *Xerula strigosa* has previously been reported only in China and Pakistan (*Wang et al., 2008*; *Krisai-Greilhuber et al., 2017*). In our phylogenetic analysis, *X. strigosa* is sister with *X. sinopudens* (Fig. 2), which is consistent with the results of *Qin et al. (2014)*. Both specimens have small basidiomata and a pileus surface covered with setae. However, *X. strigosa* can be distinguished from *X. sinopudens* by its narrowly utriform pleurocystidia, which are capitate to subcapitate at the top and often have yellowish crystalline deposits (*Wang et al., 2008*; *Liu, Zhao & Hyde, 2009*).

## ACKNOWLEDGEMENTS

We are very thankful to Xinya Yang (Engineering Research Center of Edible and Medicinal Fungi, Ministry of Education, Jilin Agricultural University, China) for her help with this study. We are also particularly grateful to the editors and reviewers for their valuable suggestions and comments on this paper.

### Funding

The study was supported by the Jilin Province Science and Technology Development Plan Project (No. 20230202119NC), the study on the species diversity of wild economic fungal resources in Biyang County, Henan Province, China; Research on the Creation of Excellent Edible Mushroom Resources and High Quality & Efficient Ecological Cultivation Technology in Jiangxi Province (20212BBF61002) and the Scientific and Technological Tackling Plan for the Key Fields of Xinjiang Production and Construction Corps (No. 2021AB004). The funders had no role in study design, data collection and analysis, decision to publish, or preparation of the manuscript.

### Grant Disclosures

The following grant information was disclosed by the authors:
The Jilin Province Science and Technology Development Plan: 20230202119NC.
Study on the species diversity of wild economic fungal resources in Biyang County, Henan Province, China.
Research on the Creation of Excellent Edible Mushroom Resources and High Quality & Efficient Ecological Cultivation Technology in Jiangxi Province: 20212BBF61002.
The Scientific and Technological Tackling Plan for the Key Fields of Xinjiang Production and Construction Corps: 2021AB004.

### Competing Interests

The authors declare there are no competing interests.

### Author Contributions

- Ya-jie Liu conceived and designed the experiments, performed the experiments, analyzed the data, prepared figures and/or tables, authored or reviewed drafts of the article, and approved the final draft.
- Zheng-xiang Qi performed the experiments, analyzed the data, prepared figures and/or tables, and approved the final draft.
- You Li performed the experiments, prepared figures and/or tables, and approved the final draft.
- Lei Yue performed the experiments, prepared figures and/or tables, and approved the final draft.
- Gui-ping Zhao performed the experiments, prepared figures and/or tables, and approved the final draft.

- Xin-yue Gui performed the experiments, analyzed the data, prepared figures and/or tables, and approved the final draft.
- Peng Dong performed the experiments, prepared figures and/or tables, and approved the final draft.
- Yang Wang analyzed the data, prepared figures and/or tables, authored or reviewed drafts of the article, and approved the final draft.
- Bo Zhang conceived and designed the experiments, authored or reviewed drafts of the article, and approved the final draft.
- Xiao Li conceived and designed the experiments, authored or reviewed drafts of the article, and approved the final draft.

## DNA Deposition

The following information was supplied regarding the deposition of DNA sequences:

The ITS sequences are available at NCBI: OR035776, OR035770, OR035771, OR035772, OR035773, OR035774, OR035775, OR035766, OR035767, OR035768, OR035769, OR030918.

The nrLSU sequences are available at NCBI: OR036095, OR036089, OR036090, OR036091, OR036092, OR036093, OR036094, OR036085, OR036086, OR036087, OR036088, OR030419.

## Data Availability

The multigene matrix of Hymenopellis and Xerula are available in the Supplementary Files.

## New Species Registration

The following information was supplied regarding the registration of a newly described species:

Hymenopellis biyangensis Y.J. Liu, B. Zhang & Xiao Li sp. nov.

MycoBank No: 848957

## Supplemental Information

Supplemental information for this article can be found online at http://dx.doi.org/10.7717/peerj.16681#supplemental-information.

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
