# Peer review of "A new species and new records of Hymenopellis and Xerula (Agaricales, Physalacriaceae) from China"

_PeerJ, doi:10.7717/peerj.16681_

## Round 0.1 · original submission · Major Revisions

Two experts in this field assessed your manuscript and agreed that it contains relevant information but requires significant work to improve presentation and readability. Improvement of the descriptions and observations of the species and well-described methodologies have to be included in a revised version of the manuscript.

·

Basic reporting

Language and grammar
The English language is good, there are few grammatical errors in some paragraphs. I marked some examples in the PDF file.
Literature and background
The information provided in the introduction is adequate, however, I believe that some paragraphs should be revised and rewritten. In addition, the objective of the article needs to be organized. The introduction has the necessary information to understand the work relevance.
Regarding the discussions, these do explain the results. The findings are discussed and it is clear why they were performed. In addition, literature that is linked to the results is included. Interesting aspects of the results are included in the discussion.
Structure, figures, and tables
The structure of the article is good
Regarding the figures, many of the titles and descriptions are not accurate or are incomplete and need to be corrected. These are marked on the PDF files.

Experimental design

Research question
The research question is well defined, also it is congruent to what is concluded.
Rigorous investigation
The approaches and methodologies are appropriate to answer the initial question. Some corrections are needed in the materials and methods section. Comments can be found in the PDF file.
Methods
Some methodologies need to be complemented; details are missing, which makes it impossible to reproduce the technique. These methods are marked on the PDF.

Validity of the findings

Underlying data
The respective controls are included in the work.
Conclusions
This work does not present conclusions.

Additional comments

In general, the research question and methodologies are interesting and appropriate, as well as the results obtained. I consider it necessary to improve the quality of the figures and the descriptions of each one. In addition, in the section of materials and methods they should indicate what the abbreviations mean. The results are correctly explained, and it is clear for the reader to understand why they are new species.

Reviewer 2 ·

Basic reporting

The theoretical framework is well developed, it includes all the necessary works that are relevant to the subject. However, in some parts of the introduction the information presented is not clear, requiring rewriting. Some observations are made in the manuscript.
The structure of the manuscript has some deficiencies, information is mixed between M&M and Results (they are marked in the manuscript).
The photographs and illustrations are of good quality, except for a few exceptions that are not well represented (ex. Fig 5e).
Many references in the text are miscited. The references have many deficiencies (see attached). The correspondence between what was cited in the text and the bibliographical references and viceversa was not checked.

Experimental design

The methodology is the traditionally used for this type of study. It's correct. Molecular analyzes (sequencing, amplification) are superficially explained, it is recommended to expand that section.
In the morphological analyses, a specific criteria for the description of the structures is not followed, which is recommended (ex. Largent, Vellinga or another).

Validity of the findings

The descriptions are very concise and apparently with very limited observations, it is necessary to describe the observed structures with more variability. It is recommended that authors follow the criteria of Petersen & Hughes 2010 to describe the species. Many missing data in the descriptions are essential to be able to corroborate if the described species is well identified or if the proposed species really is. The order of the descriptions is also not the one usually used for this group of organisms, making it a bit more difficult to quickly find the description of any particular structure, although this is not a major problem.

The diagnostic characters of the species are not correctly defined. For example for the new species proposed: “…slightly raised in the center, and a nearly radial distribution of folds, cheilocystidia are abundant and form sterile bands. The pileipellis consists of sphaeropedunculate and clavate cells” are characters common to almost all species of Hymenopellis. Basidia 2-sporic is another character shared by many species, including H. amygdaliformis f. bispora, the most closely related species according to the authors.

Observations in all species require further elaboration. Comparisons with species are incomplete. Species that deserve to be compared are not compared, from a morphoanatomical, phylogenetic or biogeographical perspective, especially in the proposed new species. With the data provided for the authors I do not have the tools to verify if what is described is a new species, to whom it is related, or if it is a species already described.

Additional comments

The manuscript is interesting and could be published in this or another journal, but it requires more work, improving the descriptions and observations of the species. I consider rejecting the manuscript because it is impossible for me to corroborate with the data provided if the proposed species is really a new species. I hope the authors can complete the manuscript and submit it again.

Annotated reviews are not available for download in order to protect the identity of reviewers who chose to remain anonymous.

---

## Round 0.2 · Major Revisions

One Reviewer still thinks there is a great opportunity area in the description of results. Please address these comments.

·

Basic reporting

I have no comments for this section. All revisions were taken into account by the researchers.

Experimental design

I have no comments for this section. All revisions were taken into account by the researchers.

Validity of the findings

I have no comments for this section. All revisions were taken into account by the researchers.

Additional comments

All revisions made to this work were taken into account. I consider that this version of the article can be published.

Reviewer 2 ·

Basic reporting

The structure of the manuscript improved, there are still some errors in the biblography (less than before). The theoretical framework is correct and the results are relevant for publication.

Experimental design

The methodology is the traditionally used for this type of study. It's correct. In the morphological analyses, a specific criteria for the description of the structures is not followed, which is recommended (ex. Largent, Vellinga or another).

Validity of the findings

The descriptions are very concise and apparently with very limited observations, it is necessary to describe the observed structures with more variability. It is recommended that authors follow the criteria of Petersen & Hughes 2010 to describe the species. Many missing data in the descriptions are essential to be able to corroborate if the described species is well identified or if the proposed species really is. The order of the descriptions is also not the one usually used for this group of organisms, making it a bit more difficult to quickly find the description of any particular structure, although this is not a major problem.

The diagnostic characters of the species are not correctly defined. For example for the new species proposed: “…slightly raised in the center, and a nearly radial distribution of folds, cheilocystidia are abundant and form sterile bands. The pileipellis consists of sphaeropedunculate and clavate cells” are characters common to almost all species of Hymenopellis. Basidia 2-sporic is another character shared by many species, including H. amygdaliformis f. bispora, the most closely related species according to the authors.

Observations in all species require further elaboration. Comparisons with species are incomplete. Species that deserve to be compared are not compared, from a morphoanatomical, phylogenetic or biogeographical perspective, especially in the proposed new species. With the data provided for the authors I do not have the tools to verify if what is described is a new species, to whom it is related, or if it is a species already described.

Additional comments

The manuscript could be published after severe corrections. The description of the species in the distribution extensions are correct, but it is recommended to improve (expand) the description of the proposed new species. Many characters of taxonomic importance for the genus are not well explained in the description, and it generates confusion between species. The structures that are not described (ex. Pileocystidia) are not known if they were not observed, or really do not exist in the specimen. Furthermore, assuming that these structures are absent, because of the different identification keys of the genus, I consider that the proposed species is more similar to other species compared by the authors.
I hope the authors can complete the manuscript and submit it again. but to pay special attention to the description of the proposed new species.

Annotated reviews are not available for download in order to protect the identity of reviewers who chose to remain anonymous.

---

## Round 0.3 · Minor Revisions

I would like to thank the authors for this revised version of the manuscript. The only pending issue to address is related to English usage. Please seek the advice of a professional proofreading service.

**Language Note:** The Academic Editor has identified that the English language must be improved. PeerJ can provide language editing services - please contact us at copyediting@peerj.com for pricing (be sure to provide your manuscript number and title). Alternatively, you should make your own arrangements to improve the language quality and provide details in your response letter. – PeerJ Staff

Reviewer 2 ·

Basic reporting

I consider it essential that the article be evaluated by a fluent English speaker.
In addition to that I don´t have new comments for the manuscript.

Experimental design

Some structures are misinterpreted in the descriptions; they consider pileocystidia to be those that are actually elements of a hymenoderm or ixohymenoderm. Authors should correct this in all descriptions.

Validity of the findings

I don´t have new comments for the manuscript.

Additional comments

The manuscript with comments is attached.
I think it is essential that the article be evaluated by a native English speaker.

Annotated reviews are not available for download in order to protect the identity of reviewers who chose to remain anonymous.

---

## Round 0.4 · accepted · Accept

Thanks for the improvement in the manuscript.